# Oral Administration of Probiotic *Bifidobacterium breve* Improves Facilitation of Hippocampal Memory Extinction via Restoration of Aberrant Higher Induction of Neuropsin in an MPTP-Induced Mouse Model of Parkinson’s Disease

**DOI:** 10.3390/biomedicines9020167

**Published:** 2021-02-08

**Authors:** Toshiaki Ishii, Hidefumi Furuoka, Motohiro Kaya, Tetsuya Kuhara

**Affiliations:** 1Department of Basic Veterinary Medicine, Obihiro University of Agriculture and Veterinary Medicine, Obihiro 080-8555, Hokkaido, Japan; furuoka@obihiro.ac.jp; 2Center for Regional Collaboration in Research and Education, Obihiro University of Agriculture and Veterinary Medicine, Obihiro 080-8555, Hokkaido, Japan; mkaya@obihiro.ac.jp; 3Next Generation Science Institute, Morinaga Milk Industry Co., Ltd., Zama 252-8583, Kanagawa, Japan; t_kuhara@morinagamilk.co.jp

**Keywords:** Parkinson’s disease, MPTP, *Bifidobacterium breve* strain A1 [MCC1274] (*B. breve* A1), memory extinction, dopamine, hippocampus, neuropsin, postsynaptic density protein-95 (PSD95), synaptophysin (SYP), spine, substantia nigra pars compacta (SNpc)

## Abstract

We previously reported that 1-methyl-4-phenyl-1,2,3,6-tetrahydropyridine-induced Parkinson’s disease (PD) model mice (PD mice) facilitate hippocampal memory extinction, which may be the cause of cognitive impairment in PD. Recent studies on the consumption of probiotics have reported a variety of beneficial effects on the central nervous system via the microbiota–gut–brain axis. In this study, we investigated the effects of oral administration of *Bifidobacterium breve* strain A1 [MCC1274] (*B. breve* A1) on the facilitation of hippocampal memory extinction observed in PD mice. We found that four-day consecutive oral administration of *B. breve* A1 restored facilitation of contextual fear extinction in PD mice. Hippocampal mRNA expression levels of postsynaptic density protein-95 and synaptophysin significantly decreased in the PD mice, but mRNA and protein expression levels of neuropsin increased. Furthermore, CA1 apical spine density was significantly reduced in PD mice. On the other hand, administration of *B. breve* A1 to PD mice recovered all these expression levels and the CA1 spine density to control levels. These results suggest that increased induction of neuropsin is involved in abnormal changes in hippocampal synaptic plasticity, and that *B. breve* A1 imposes reins on its expression, resulting in the restoration of abnormal hippocampal synaptic plasticity and the facilitation of fear extinction in PD mice.

## 1. Introduction

Parkinson’s disease (PD) is the neurodegenerative disease caused by the selective and progressive loss of dopaminergic (DAergic) neurons in the substantia nigra pars compacta (SNpc), which leads to a reduction in DAergic fibers in the striatum from the nucleus and results in extrapyramidal motor dysfunctions in patients with PD [1]. Non-motor symptoms sometimes appear in patients with PD before the onset of motor dysfunction [2,3]. Particularly, the probability of onset of cognitive impairment is relatively high, appearing in about 40% of PD patients [4].

It recently became widely known that probiotics are good for health through the daily intake of adequate amounts of living microorganisms [5]. Indeed, several species of probiotics have been reported to maintain a suitable intestinal environment [6], reduce the risk of infectious disease via normalization of the immune system [7,8], and help prevent the onset of obesity [9] and cancer [10] through improving the intestinal bacterial flora. Furthermore, it has been reported that the application of some probiotics is beneficial for the central nervous system via the microbiota–gut–brain axis [11,12,13]. For example, administration of **Bifidobacterium* breve* strain A1 [MCC1274] (*B. breve* A1) was shown to prevent cognitive impairment in Alzheimer’s disease (AD) model mice [14], daily administration of probiotic formulation SLAB51 was found to restore behavioral impairment and rescue DAergic neurons in SNpc and striatum in 6-hydroxydopamine-induced PD model mice [15], administration of *Lactobacillus plantarum* PS128 was found to improve anxiety-like behaviors in mice exposed to early life stress [16], and administration of VSL#3, a probiotic mixture of eight different gram-positive bacterial species, were found to modulate neuronal activities such as long-term potentiation in young and aged rats [17].

Recently, we reported that hippocampal memory extinction was facilitated in 1-methyl-4-phenyl-1,2,3,6-tetrahydropyridine (MPTP)-induced PD model mice compared with control mice, the facilitation of which may be involved in cognitive impairment in PD [18]. With the overall aim of exploring the potential pharmaceutical use of *B. breve* A1 for the treatment of cognitive deficits in PD, in the present study, we examined the effects of *B. breve* A1 on contextual fear extinction to clarify whether administration of *B. breve* A1 can restore the facilitation of hippocampal memory extinction in PD mice.

## 2. Materials and Methods

### 2.1. Animals and MPTP Treatment

Male C57BL/6 mice (7–8 weeks old) (Ishibe breeding facility, Clea Japan, Tokyo, Japan) were maintained under controlled temperature (22 ± 2 °C) and humidity (35 ± 5%) on a 12-h light/12-h dark cycle (lights on at 07:00) and allowed ad libitum access to pellet food (Clea Japan, Tokyo, Japan) and water. All procedures for the care and use of experimental animals were approved by the Animal Research Committee at the Obihiro University of Agriculture and Veterinary Medicine *(No. 20-1, 1 April 2020)* and conducted in compliance with the 1989 Guiding Principles for the Use of Animals in Toxicology.

MPTP (Sigma-Aldrich, Tokyo, Japan)-treated PD model mice were prepared as described in our previous report [17,18,19,20]. Eight-week-old mice were given four intraperitoneal injections of a single dose of 20 mg/kg (in 100 µL) in the first two injections and 15 mg/kg (in 100 µL) in the last two injections every 2 h. Saline was administered in a similar manner as a control. Subsequent experiments were conducted 7 days after the last MPTP or saline injection. Each of MPTP-treated (*n* = 73) and saline-treated mice (*n* = 73) was divided into the following three groups: *Control-Saline*, *n* = 36; Control-*B. breve* A1, *n* = 32; Control-Non-viable *B. breve* A1, *n* = 5; MPTP-Saline, *n* = 36; MPTP-*B. breve* A1, *n* = 32; MPTP-Non-viable *B. breve* A1, *n* = 5.

### 2.2. Contextual Fear Conditioning Test

All procedures were performed as described in our previous report [18]. The experiments consisted of three phases: conditioning, training, and testing. All experiments were conducted in a conditioning chamber (17.0 × 17.0 × 14.5 cm) with a stainless-steel rod floor through which electrical foot shocks could be delivered (ST-10; Melquest, Toyama, Japan). All behaviors of mice in the chamber were monitored and recorded by an overhead color CCD camera. During each of these phases, freezing behavior was defined as a complete absence of movement lasting for longer than 1 s, except for respiration and heartbeat. The percentage of time spent freezing was measured during the test session (time spent freezing/total time × 100) as an index of memory. During the conditioning phase, as the conditioned stimulus (CS), mice were placed in the chamber, and as the unconditioned stimulus (US), two-foot shocks were delivered (2-s duration, 2 mA). Mice were returned to their home cage 30 s after the final foot shock. Next, mice underwent extinction training twice every 24 h after conditioning. In this training, mice were re-exposed to the CS for 30 min without any US, and the percentage of time spent freezing during the initial 3 min of the entire duration of exposure was assessed. Then, 24 h after the last extinction training, the mice were re-exposed to the CS for 3 min, and the percentage of time spent freezing was assessed (extinction test).

### 2.3. cAMP Assay

cAMP levels were determined using a Cyclic AMP EIA kit (Cayman Chemical Co., Ann Arbor, MI, USA). All procedures were performed as described in our previous report [18]. The hippocampus was collected immediately after the extinction test (Day 3) (Figure 1) and then homogenized in 10 volumes of 5% trichloroacetic acid (TCA) in water on ice using a polytron-type homogenizer (*Microtec* Co., Chiba, Japan). After centrifugation of the sample at 1500× *g* for 10 min, the supernatant was collected. To extract TCA from the sample, four volumes of water-saturated ether was added to one volume of the supernatant and vigorously vortexed. After removal of the top ether layer, the same treatment was repeated another two times. After the final removal of the top ether layer, the sample was heated at 70 °C for 5 min to remove the residual ether. The samples were subjected to an enzyme-linked immunosorbent assay according to the manufacturer’s protocol. The absorbance was measured at a wavelength of 415 nm. The tissue pellet precipitated after the TCA extraction was washed once with an ethanol/ether (1:3) solution, heated at 70 °C for 3 min to remove the residual ethanol/ether, and then weighed. cAMP levels are shown as pmol/mg of the weight of TCA-precipitated tissue protein.

### 2.4. RNA Extraction and Real-Time Quantitative Polymerase Chain Reaction (RT-qPCR) Assay

Total RNA was extracted from hippocampal tissue using Direct-zol™ RNA MiniPrep (Zymo Research, Tustin, CA, USA) and quantified using the QuantiFluor™ RNA system (Promega, Madison, WI, USA) according to the manufacturers’ instructions. RNA was amplified using the MyGo Mini Real-Time PCR system (IT-IS Life Science, Ltd., Cork, Ireland). One-step RT–qPCR was performed using MyGo Green 1-step Low Rox (IT-IS Life Science, Ltd.) for a total volume of 20 µL and a template concentration of 10 pg/µL total RNA, according to the manufacturer’s recommendations. The thermal cycling conditions were 45 °C for 10 min as an RT step and 95 °C for 2 min, followed by 40 cycles of 95 °C for 10 s and 60 °C for 20 s. The relative quantification (fold change) of mRNA expression was estimated by the use of the 2^−ΔΔCt^ method [21], as described in our previous report [18]. β-actin was used as the housekeeping gene for each sample to normalize the targeted gene expression. The primers used in this study are shown in Table 1.

### 2.5. Golgi–Cox Staining

Mice were transcardially perfused with ice-cold phosphate-buffered saline (PBS) under isoflurane anesthesia and then decapitated. The brain was removed and then processed with Golgi–Cox staining using the *super*Golgi kit (Bioenno Tech, LLC, Santa Ana, CA, USA) according to the manufacturer’s instructions.

After impregnation, the brains were cut into consecutive 160-µm thick sections containing the hippocampus (positioned 1.74–2.54 mm posterior to the bregma) using an oscillating tissue slicer (Linear Slicer Pro7; Dosaka, Kyoto, Japan). The sections were placed on silane-coated glass slide (New Silane III; MUTO PURE CHEMICALS, Tokyo, Japan). After staining and dehydration, the sections were mounted with Mount-quick media (DAIDO SANGYO, Tokyo, Japan). We analyzed the spine density of secondary apical dendrites in CA1 pyramidal neurons using biological light microscopy (BX-41, Olympus, Tokyo, Japan) and image capture system (DP-70 with DP manager, Olympus). Nine dendrites of nine neurons per mouse were quantified for the number of spines, and spine density was calculated as spines per 10 µm.

### 2.6. Western Blot Analysis

Hippocampal tissue was solubilized in three volumes of RIPA lysis buffer (Santa Cruz Biotechnology, Dallas, TX, USA) with protease inhibitor cocktail at 4 °C and then centrifuged at 14,000× *g* for 20 min. The supernatant was collected and the protein concentration was determined using the BCA Protein Assay Kit (Takara Bio, Shiga, Japan). After mixing with an equal volume of 2 × sodium dodecyl sulphate (SDS)-sample buffer (2% SDS, 10% glycerol, 10% 2-mercaptoethanol, 0.1% bromophenol blue, and 62.5 mm Tris-HCl, pH 6.8), equal amounts of protein were subjected to polyacrylamide gel electrophoresis (12.5% polyacrylamide) and then transferred onto nitrocellulose membranes at 4 °C in 25 mm Tris-HCl, 192 mM glycine, 20% methanol, and 0.025% SDS. After blocking with 4% Block Ace (KAC, Tokyo, Japan) dissolved in PBS containing 0.1% (*v*/*v*) Tween 20 (TBST) for 1 h at room temperature, the membranes were incubated with rat anti-neuropsin monoclonal antibody (MBL, Nagoya, Japan; 1:1000 in TBST) or rabbit anti-β-tubulin polyclonal antibody (Proteintech, Tokyo, Japan; 1:2000 in TBST) at 4 °C overnight. After incubation, the membranes were washed three times in TBST and probed with goat anti-rat IgG antibody or goat anti-Rabbit IgG conjugated to horseradish peroxidase (Cell Signaling Technology, Danvers, MA, USA; 1:3000 in TBST) at room temperature for 1 h. After washing the membranes three times in TBST, the final protein-IgG complexes were identified using Amersham ECL Western Blotting Detection reagents (Cytiva, Tokyo, Japan), followed by detection with LAS3000 (Fuji Photo Film, Tokyo, Japan). Assessment of the band intensities was performed using Multi-Gauge software (Fuji Photo Film). The protein levels were normalized to β-tubulin as a loading control.

### 2.7. Administration of B. breve A1

*B. breve* A1 is a probiotic strain stocked as strain MCC1274 in Morinaga Culture Collection, Zama, Japan. Lyophilized *B. breve* A1 was supplied by Morinaga Milk Industry Co., Ltd., and stored at 4 °C under anaerobic conditions until use. Lyophilized *B. breve* A1 was suspended in saline at a concentration of 5 × 10^9^ cfu/mL just before administration. Nonviable *B. breve* A1 was prepared by further heat-shock treatment of the suspended bacterium at 60 °C for 60 min and stored at −20 °C until use, following a previous report [14]. Mice were given daily oral administration of living or heat-killed *B. breve* A1 at a volume of 0.2 mL (1 × 10^9^ cfu organisms) for 4 days from 3 days before the conditioning day (Figure 1). For oral gavage, mice were administered using a mouse feeding stainless steel bulbous-ended needle (0.92 × 70 mm, AS ONE, Osaka, Japan) and the bulbous-ended needle was inserted over the tongue into the stomach.

### 2.8. Data Analysis

Multiple group comparisons were assessed using one-way analysis of variance, followed by Tukey’s post-hoc test or Kruskal–Wallis analysis and Mann–Whitney *U* tests based on the analyses of homoscedasticity using Levene’s test. Statistical differences were considered significant when *p* < 0.05. All statistical analyses were performed using SPSS software (version 16.0; SPSS Japan, Inc., Tokyo, Japan).

## 3. Results

### 3.1. Administration of B. breve A1 to PD Mice Improved Facilitation of Contextual Fear Extinction

Previously, we reported that MPTP-induced PD mice show facilitation of memory extinction and attenuation of memory retention without any noticeable effects on consolidation and reconsolidation in a contextual fear-conditioning test [19]. In the present study, we examined the effects of daily oral administration of *B. breve* A1 for 4 days on the facilitation of fear extinction in PD mice. Figure 1 shows that a significant decrease in freezing levels in PD mice was observed on both Day 2 and Day 3 compared with Day 1, but only on Day 3 in control mice, suggesting that memory extinction was facilitated in the PD mice. These results obtained were the same as those from our previous report [18]. Administration of *B. breve* A1 also prevented the facilitation of contextual fear extinction in PD mice on Days 2 and 3, but did not have any significant effect on contextual fear extinction in control mice (Figure 2). On the other hand, oral administration of nonviable *B. breve* A1 did not show any significant effects on the facilitation of contextual fear extinction in PD mice, showing the facilitation in a manner similar to PD mice (Figure 3).

### 3.2. Effect of B. breve A1 on the Hippocampal mRNA Expression Levels of Neuropsin, Synaptophysin (SYP), Postsynaptic Density Protein-95 (PSD95), Brain-Derived Neurotrophic Factor (BDNF), and Ionized Calcium-Binding Adapter Molecule 1 (Iba1)

Synaptic proteins play an important role in the regulation of learning and memory function. Therefore, we examined the mRNA expression levels of several proteins involved in synaptic formation, stability, and plasticity. As shown in Figure 4A, compared with the control mice, the mRNA expression level of neuropsin, a plasticity-related extracellular protease [22], in PD mice was significantly increased, while administration of *B. breve* A1 recovered its increased level to the control level. In contrast to the mRNA expression of neuropsin, in PD mice, both postsynaptic density protein-95 (PSD95) and synaptophysin (SYP) mRNA expression levels were significantly decreased, while administration of *B. breve* A1 prevented their decrease (Figure 4B,C).

On the other hand, no significant differences were seen in the levels of ionized calcium-binding adapter molecule 1 (Iba1) and brain-derived neurotrophic factor (BDNF) mRNA expression levels between the four groups (Figure 5). These results suggest that abnormal changes in hippocampal synaptic plasticity occur in PD mice in a neuropsin-dependent manner.

### 3.3. B. breve A1 Restored Decreased Dendritic Spine Density in PD Mice

Because the mRNA expression levels of both SYP and PSD95, which are markers of presynaptic and postsynaptic proteins, respectively, were significantly decreased, we next examined CA1 apical spine density in the hippocampus. Histological analysis of Golgi–Cox staining revealed that CA1 apical spine density was significantly reduced in PD compared with control mice (Figure 6), indicating decreased neuronal plasticity in PD mice. By contrast, administration of *B. breve* A1 prevented the reduction of spine density in PD mice and maintained it at the same level as that in control mice (Figure 6). On the other hand, administration of *B. breve* A1 did not alter spine density in control mice.

### 3.4. B. breve A1 Does Not Affect cAMP Levels in the Hippocampus

We previously reported that hippocampal cyclic adenosine monophosphate (cAMP) levels decrease in MPTP-treated compared with control mice [18]. Moreover, we also demonstrated that administration of the phosphodiesterase IV (PDE IV) inhibitor rolipram [20] and/or the serotonin 5-hydroxytryptamine receptor 4 receptor (5-HT_4_R) agonists prucalopride and velusetrag [18] restored facilitation of contextual fear extinction in PD mice via increasing hippocampal cAMP levels. Therefore, we examined whether *B. breve* A1 could increase hippocampal cAMP levels. Figure 7 shows that hippocampal cAMP levels were significantly decreased in PD compared with control mice. However, administration of *B. breve* A1 did not show any significant effects on hippocampal cAMP levels in PD mice (Figure 7). Similarly, *B. breve* A1 did not have any effects on the level in control mice. These results suggest that *B. breve* A1 improves facilitation of contextual fear extinction in PD mice, but not through the activation of the cAMP-dependent protein kinase signaling pathway in the hippocampus.

### 3.5. Effect of B. breve A1 on Neuropsin Protein Expression in the Hippocampus

Western blot analysis showed that the protein expression level of neuropsin was significantly increased in PD compared with control mice (Figure 8). On the other hand, administration of *B. breve* A1 prevented an increase in the neuropsin expression level in PD mice and maintained it at the same level as that in the control mice (Figure 8). No significant differences were found between the control mice with and without administration of *B. breve* A1.

## 4. Discussion

The results of this study demonstrated that oral administration of *B. breve* A1 restores the facilitation of contextual fear extinction in PD mice via the prevention of abnormal changes in hippocampal synaptic plasticity, the mechanism of which is probably due to the normalizing of an aberrant higher induction of neuropsin without stimulating the cAMP/cAMP response element-binding protein (CREB) pathway in the hippocampus.

It has been reported that neurodegenerative disorders, which lead to the onset of cognitive impairments such as AD, cause dendritic spine alterations in terms of number and/or structure, as well as the function of these postsynaptic sites [23,24]. Because the structure and functional changes in the synapses of hippocampal neurons are accompanied by the induction or suppression of specific genes that encode synaptic proteins and extracellular enzymes implicated in synaptic plasticity [25,26], we determined the mRNA expression levels of two synaptic proteins, SYP and PSD95, in the hippocampus. The results indicated significant decreases in the mRNA expression levels of both SYP and PSD95 in PD mice; however, administration of *B. breve* A1 significantly recovered both mRNA expression levels to the control levels. Moreover, the mRNA expression level of neuropsin, a plasticity-related extracellular protease [27], was significantly increased in PD compared with control mice, while administration of *B. breve* A1 recovered the increased expression level to the control level. The same results were obtained by western blotting analysis; the protein expression level of neuropsin was significantly increased in PD compared with control mice, while administration of *B. breve* A1 recovered the increased expression level to the control level. To observe the existence of hippocampal synaptic alteration in PD mice more concretely, we conducted a histological analysis of Golgi–Cox staining. The results revealed that CA1 apical spine density was significantly reduced in PD compared with control mice. Intriguingly, administration of *B. breve* A1 prevented the reduction of spine density in PD mice and maintained it at the same level as that in the control mice. These results suggest that increased neuropsin expression might be critically involved in abnormal changes in hippocampal synaptic plasticity in PD mice, and that *B. breve* A1 can prevent the progression of behavioral and anatomical abnormal indexes by restoring the increased expression level of its protein to the control level.

Next, we examined the mRNA expression levels of BDNF and the microglial marker Iba1. Because BDNF stimulates a learning dependent-increase in the levels of synaptic structure proteins such as SYP and PSD-95, BDNF is considered to be a regulator of synaptic plasticity [28]. On the other hand, microglia are also involved in synaptic homeostasis by regulating dendritic spine formation via synaptic pruning other than the brain–immune system [29]. However, no significant changes in the mRNA expression of BDNF or Iba1 were observed in PD mice, regardless of whether it was before or after administration of *B. breve* A1. These results suggest that BDNF might not be necessary for *B. breve* A1-induced restoration of PSD95 and SYP mRNA expression levels in PD mice. Moreover, the microglia do not seem to be critically involved in the reduction of spine density in PD mice because they do not alter Iba1 mRNA expression levels.

Previously, we reported that reduced cAMP/CREB signaling in the hippocampus leads to the facilitation of memory extinction in PD mice because the administration of rolipram, a PDE IV inhibitor, restored the facilitation of memory extinction by stimulating the cAMP/CREB pathway in the hippocampus [20]. Furthermore, 5-HT_4_R agonists restored the facilitation of contextual fear extinction in PD mice by stimulating the cAMP/CREB pathway in the hippocampal dentate gyrus, the area that is innervated by the serotonergic neurons that form the final synaptic ending in a candidate neuronal pathway from the substantia nigra (SN) to the hippocampus [18]. Therefore, we examined the effect of *B. breve* A1 on hippocampal cAMP levels. However, we found that *B. breve* A1 did not increase hippocampal cAMP levels in PD mice, despite the fact that 5-HT_4_R agonists and rolipram did (Figure 7 and [18,20]). These results suggest that *B. breve* A1 restores the facilitation of contextual fear extinction in PD mice without stimulating the cAMP/CREB pathway in the hippocampus, that is, via a mechanism that differs from rolipram and 5-HT_4_R agonists. Previously, we proposed that cognitive impairment in PD mice may be caused by deficient neuronal input from the SN to the hippocampus via serotonergic neurons in the median raphe nucleus, leading to the inhibition of the hippocampal cAMP/CREB pathway [18]. However, *B. breve* A1 does not appear to compensate this neuronal pathway directly, but rather, to activate other undefined signaling pathways to prevent abnormal changes in hippocampal synaptic plasticity.

It is considered that the luminal gut microbiota could be in bidirectional communication with the brain through multiple signaling pathways derived from the neuronal, endocrine, and immune systems, as well as the nutrients and by-products of microbial metabolism [30]. Moreover, perturbation of this microbiota–gut–brain axis has been known to be involved in pathophysiology and neurodegenerative disorders [31]. Kobayashi et al. [14] reported that administration of nonviable *B. breve* A1 to AD mice, which was less effective than viable *B. breve* A1, partially ameliorated the cognitive decline observed in AD mice. On the other hand, in the present study, administration of nonviable *B. breve* A1 did not exert any significant effects on the facilitation of contextual fear extinction in PD mice (Figure 3). These results suggest that the mechanisms underlying the positive effects of *B. breve* A1 on cognitive deficits might differ between AD and PD mice. Although it remains unclear how the luminal gut *B. breve* A1 influence brain function, at least living *B. breve* A1 in the gut lumen is essential for being able to transfer its unknown signaling information to the brain. Recently, Han et al. [32] reported the existence of a vagal-to-brain axis as an integral component of the neuronal reward pathway. Their report showed that optical activation of gut-innervating vagal sensory neurons induced dopamine release from the SN via activation of specifically right, but not left, vagal sensory ganglions and, at the same time, sustained self-stimulation behavior [32]. We produced PD mice by the systemic administration of MPTP, which causes selective loss of DAergic neurons in the SNpc, but not all the DAergic neurons are destroyed, and ~40% of the neurons remain alive in the PD mice [19]. Therefore, luminal gut *B. breve* A1 may transmit to the SNpc via gut-innervating vagal sensory neurons and activate DAergic neurons in the SNpc, resulting in the prevention of abnormal changes in hippocampal synaptic plasticity via a still unknown neuronal pathway from the SNpc.

Neuropsin, which is highly expressed in the hippocampus [33], has been reported to govern synaptic plasticity through the activity-induced proteolytic cleavage of synaptic proteins [22,34] (Figure 9). Hippocampal overexpression of neuropsin via viral vectors in mice cause increased impairment in depression-like behavior and spatial memory induced by corticosterone exposure, whereas knockdown of neuropsin attenuates these impaired behaviors [35]. Moreover, the mRNA expression levels of neuropsin have been reported to be increased in the hippocampus of patients with AD [36]. Therefore, abnormal expression levels of neuropsin in the hippocampus could cause various neuronal disorders, such as depression and cognitive impairment, and it is likely that the aberrant higher induction of neuropsin in PD mice led to abnormal changes in hippocampal synaptic plasticity, resulting in impairments in memory extinction and retention. In the present study, we found that *B. breve* A1 restored the facilitation of contextual fear extinction, recovered the reduction of mRNA expression levels of two synaptic proteins, SYP and PSD95, and prevented the reduction of spine density. However, we have not yet tested the hippocampal synaptic function itself. Therefore, we need a further study to examine the effects of *B. breve* A1 on the long-term potentiation and the long-term depression, those of which are considered as neurophysiological substrates for learning and memory [37], at the Schaffer collateral–CA1 synapse in the hippocampus of PD mice.

Overall, our results indicate that oral administration of *B. breve* A1 restores the facilitation of contextual fear extinction in PD mice via the prevention of abnormal changes in hippocampal synaptic plasticity, which is likely due to the normalization of an aberrant higher induction of neuropsin (Figure 9). However, the mechanism underlying the *B. breve* A1-induced restoration of an aberrant higher induction of neuropsin in PD mice remains unknown. Further studies are needed to clarify the mechanism of the preventive effect of *B. breve* A1 on aberrant neuropsin induction and the signaling pathway of living *B. breve* A1 via the microbiota–gut–brain axis, which consequently conveys beneficial information to the hippocampus.

In conclusion, our results strongly suggest that *B. breve* A1 could be useful as a therapeutic probiotic for the prevention and treatment of cognitive deficits in PD because *Bifidobacterium* in general is known to be a highly safe probiotic bacterium and can be ingested daily over the long term. We hope that these findings can contribute to the development of a novel therapeutic method against cognitive deficits in PD.

## 5. Conclusions

We found that oral administration of *B. breve* A1 restored the facilitation of contextual fear extinction in PD mice through the restoration of an aberrant higher induction of neuropsin. Significant decreases in the mRNA expression levels of both SYP and PSD95 were observed in PD mice, but administration of *B. breve* A1 significantly recovered these to the control levels. Moreover, CA1 apical spine density was significantly reduced in PD compared with control mice, but administration of *B. breve* A1 prevented the reduction of spine density in PD mice and maintained it at the same level as that in control mice. These findings suggest that *B. breve* A1 could be useful as therapeutic probiotic for the prevention and treatment of cognitive deficits in PD by improving abnormal hippocampal CA1 synaptic plasticity.

## Figures and Tables

**Figure 1 biomedicines-09-00167-f001:**
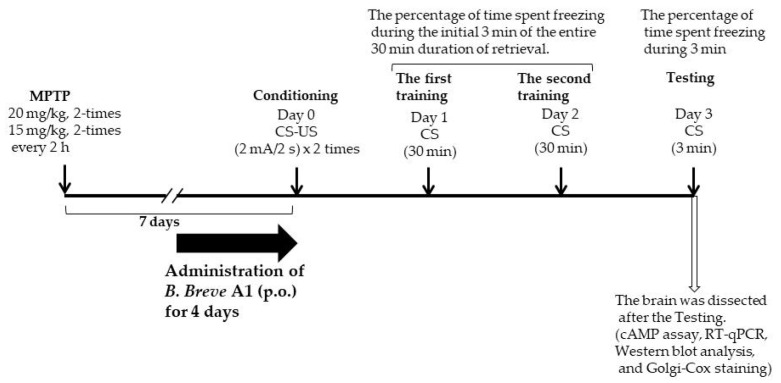
A schematic figure explaining the planning of the experiments.

**Figure 2 biomedicines-09-00167-f002:**
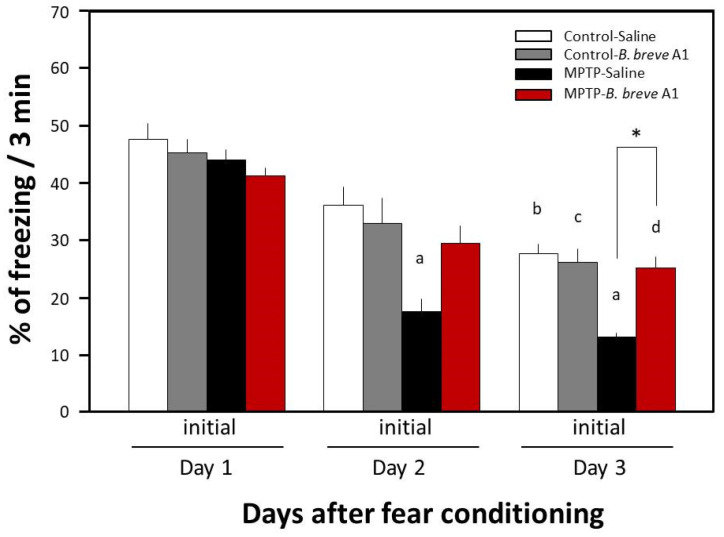
Effects of *B. breve* A1 on contextual fear extinction in mice. Mice were given oral administration of living *B. breve* A1 at a volume of 0.2 mL (1 × 10^9^ cfu organisms suspended in saline) or saline for 4 days from 3 days before the conditioning day. Data are expressed as the mean ± SEM; *n* = 6–7. ^a^
*p* < 0.05 vs. MPTP + Saline on Day 1, ^b^
*p* < 0.05 vs. Control + Saline on Day 1, ^c^
*p* < 0.05 vs. Control + *B. breve* A1 on Day 1, ^d^
*p* < 0.05 vs. MPTP + *B. breve* A1 on Day 1, and * *p* < 0.05: between MPTP + Saline and MPTP + *B. breve* A1 on Day 3.

**Figure 3 biomedicines-09-00167-f003:**
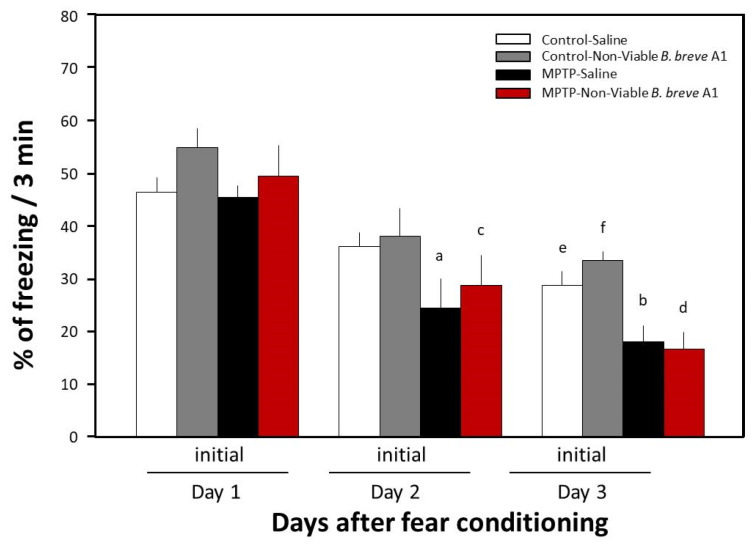
Effects of nonviable *B. breve* A1 on contextual fear extinction in mice. Mice were given oral administration of nonviable *B. breve* A1 at a volume of 0.2 mL (1 × 10^9^ cfu organisms suspended in saline) or saline for 4 days from 3 days before the conditioning day. Data are expressed as the mean ± SEM; *n* = 5. ^a^
*p* < 0.05 and ^b^
*p* < 0.01 vs. MPTP + Saline on Day 1, ^c^
*p* < 0.05 and ^d^
*p* < 0.01 vs. MPTP + nonviable *B. breve* A1 on Day 1, ^e^
*p* < 0.05 vs. Control + Saline on Day 1 and ^f^
*p* < 0.05 vs. Control + nonviable *B. breve* A1 on Day 1.

**Figure 4 biomedicines-09-00167-f004:**
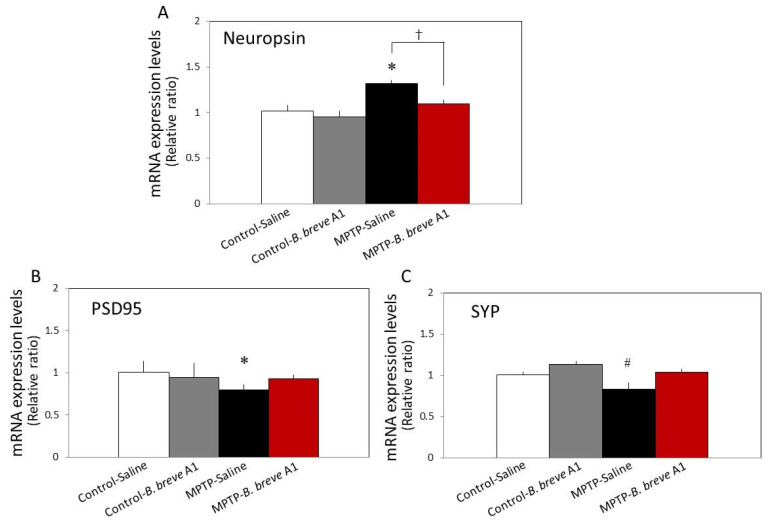
RT–qPCR analysis of the hippocampal mRNA expression levels of neuropsin (**A**), PSD95 (**B**), and SYP (**C**) in saline-administered controls and PD mice, and in *B. breve* A1-administered controls and PD mice. Data are expressed as the mean ± SEM: *n* = 7–9 per group. * *p* < 0.01 and ^#^
*p* < 0.05 vs. Control + saline and ^†^
*p* < 0.05: between MPTP + Saline and MPTP + *B. breve* A1.

**Figure 5 biomedicines-09-00167-f005:**
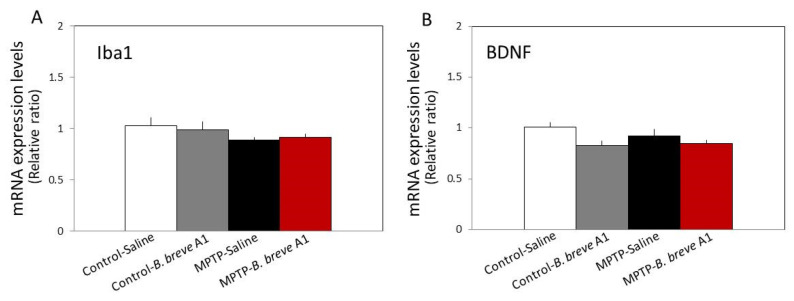
RT–qPCR analysis of the hippocampal mRNA expression levels of Iba1 (**A**) and BDNF (**B**) in saline-administered controls and PD mice, and in *B. breve* A1-administered controls and PD mice. Data are expressed as the mean ± SEM: *n* = 8–10 per group. No significant differences were observed between groups.

**Figure 6 biomedicines-09-00167-f006:**
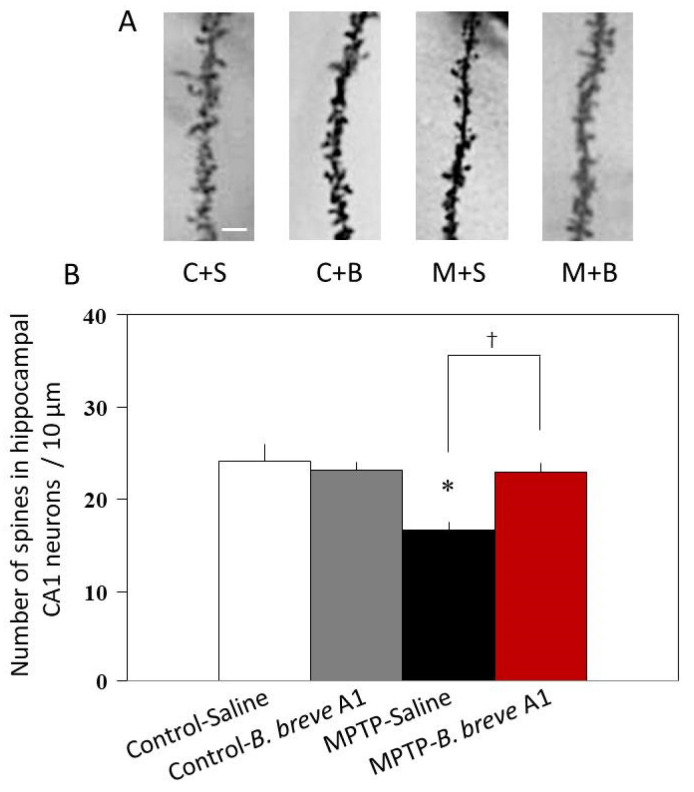
Administration of *B. breve* A1 restored the decreased dendritic spine density in the CA1 region of the hippocampus. (**A**) Representative images of spines. Scale bar = 2 µm. (**B**) The dendritic spine density per 10 µm decreased in PD mice. Administration of *B. breve* A1 prevented the reduction of spine density in PD mice. Data are expressed as the mean ± SEM; *n* = 4 per group. * *p* < 0.01 vs. Control + Saline and ^†^
*p* < 0.05: between MPTP + Saline and MPTP + *B. breve* A1.

**Figure 7 biomedicines-09-00167-f007:**
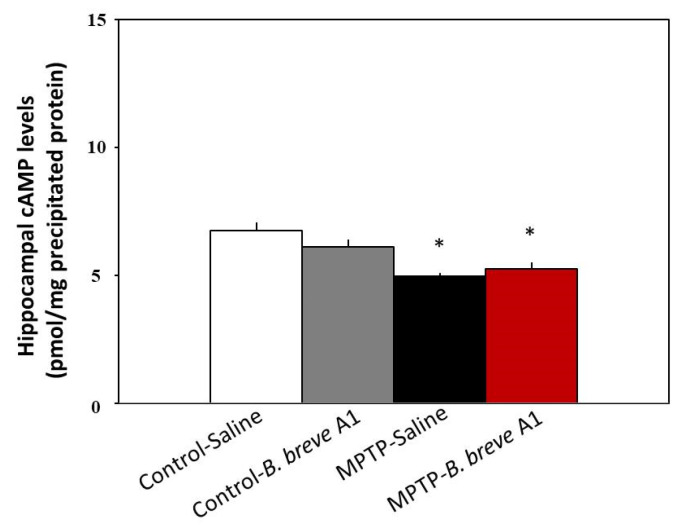
Administration of *B. breve* A1 to PD mice did not restore the decreased cAMP levels in the hippocampus. cAMP levels are shown as pmol/mg of TCA-precipitated tissue protein. Data are expressed as the mean ± SEM; *n* = 5–6. * *p* < 0.01 vs. Control + Saline.

**Figure 8 biomedicines-09-00167-f008:**
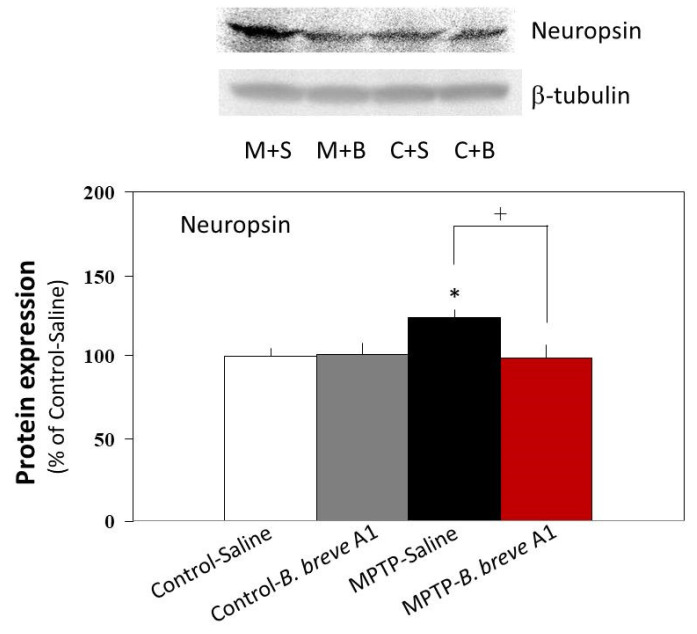
Neuropsin protein expression levels in saline-administered controls and PD mice, and in *B. breve* A1-administered controls and PD mice. The images show representative results (upper: neuropsin, lower: β-tubulin). The protein levels were normalized to β-tubulin as a loading control. Results are shown as a percentage of protein expression levels in Control + Saline. Data are expressed as the mean ± SEM: *n* = 5 per group. * *p* < 0.05 vs. Control + Saline and ^†^
*p* < 0.05: between MPTP + Saline and MPTP + *B. breve* A1.

**Figure 9 biomedicines-09-00167-f009:**
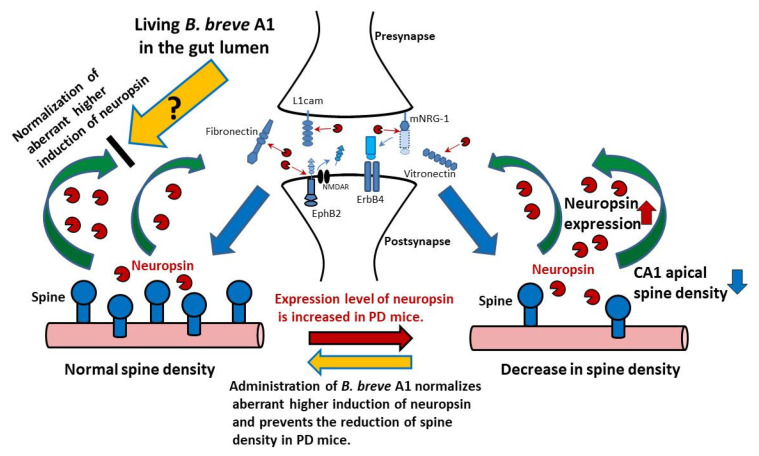
Living *B. breve* A1 in the gut lumen leads to the restoration of PD-induced aberrant higher induction of neuropsin, the mechanism of which is caused by a still unknown signal via the gut–brain axis. Neuropsin-mediated signaling plays an important role in synaptic plasticity; physiologically activated neuropsin cleaves extracellular proteins in the synaptic cleft such as fibronectin, vitronectin, L1cam, EphB2 (the cleavage of EphB2 reduces EphB2-NMDA binding), and mNRG-1(the released moiety cleaved from mNRG-1 binds ErbB4 and triggers its phosphorylation), and modifies pre- and postsynaptic interactions and synaptic signal transmission [22,34]. However, an aberrant higher induction of neuropsin observed in PD mice causes abnormal changes in hippocampal synaptic plasticity and function. Oral administration of *B. breve* A1 restored the facilitation of contextual fear extinction and the decreased hippocampal dendritic spine density in PD mice via normalization of an aberrant higher induction of neuropsin. Abbreviations: EphB2, EPH receptor B2; ErbB4, NRG-1 receptor p185; L1cam, L1 cell adhesion molecule; mNRG-1, mature neuregulin-1; NMDAR, *N*-methyl-d-aspartic acid receptor.

**Table 1 biomedicines-09-00167-t001:** RT-qPCR primers in 5′–3′ direction.

Transcript	Primers	
**β-actin**	Sense Antisense	ATTGCTGACAGGATGCAGAAG TAGAAGCACTTGCGGTGCACG
**Neuropsin**	Sense Antisense	CTCAACTGTGCGGSSGTGAA ACTCCAGGTTTCTCGGGTTT
**PSD95**	Sense Antisense	TGTAATCCTGAAGCCCTGTC GGTTTCCGATGAAGTCCC
**SYP**	Sense Antisense	TGGAGTGTGCCAACAAGAC AGCCACGGTGACAAAGAA
**BDNF**	Sense Antisense	GCGGCAGATAAAAAGACTGC CTTATGAATCGCCAGCCAAT
**Iba-1**	Sense Antisense	GAAGCGAATGCTGGAGAAAC GACCAGTTGGCCTCTTGTGT

## Data Availability

The data presented in this study are available on request from the corresponding author.

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
