# Peer review of "Oral Administration of Probiotic Bifidobacterium breve Improves Facilitation of Hippocampal Memory Extinction via Restoration of Aberrant Higher Induction of Neuropsin in an MPTP-Induced Mouse Model of Parkinson’s Disease"

_biomedicines, 2021, doi:10.3390/biomedicines9020167_

Round 1
Reviewer 1 Report
Dear authors,
This is an interesting read. Apart from few syntax formatting, I find your paper ready for publications. Congrats!
Author Response
Responses to reviewer #1 comment
Comment: This is an interesting read. Apart from few syntax formatting, I find your paper ready for publications. Congrats!
Response: We are grateful to have your consideration. Thank you very much.
Reviewer 2 Report
Overall, the manuscript is well written, the findings are important and interesting, the methodology is adequately described, and the statistics is appropriate. Addressing the following issue should further improve the rigor, depth, and mechanistic insights of the study.
The authors found that B. breve A1 could prevent the decrease of mRNA levels of SYP and PSD95, restore decreased dendritic spine density. All these finding indicated that B. breve A1 prevents abnormal changes in hippocampal synaptic plasticity. However, the authors did not test the synaptic function in CA1 by recording long-term potentiation.
The mRNA levels and dendritic structural changes could only give limited information of B. breve A1's role in synaptic plasticity. In contrast, monitoring LTP in CA1 region can directly confirm the pathogenesis of MPTP induced Parkinson’s Disease and the pharmacological effect of B. breve A1. Thus, I suggest that the authors should test LTP changes in MPTP and B. breve A1 treatment groups.
Author Response
Responses to reviewer #2 comments
Point 1: The authors found that B. breve A1 could prevent the decrease of mRNA levels of SYP and PSD95, restore decreased dendritic spine density. All these finding indicated that B. breve A1 prevents abnormal changes in hippocampal synaptic plasticity. However, the authors did not test the synaptic function in CA1 by recording long-term potentiation.
The mRNA levels and dendritic structural changes could only give limited information of B. breve A1’s role in synaptic plasticity. In contrast, monitoring LTP in CA1 region can directly confirm the pathogenesis of MPTP induced Parkinson’s Disease and the pharmacological effect of B. breve A1. Thus, I suggest that the authors should test LTP changes in MPTP and B. breve A1 treatment groups.
Response: We greatly appreciate your comments and agree with your suggestion. In the present study, we found that B. breve A1 restored the facilitation of contextual fear extinction, recovered the reduction of mRNA expression levels of two synaptic proteins, SYP and PSD95, and prevented the reduction of spine density. However, we have not yet tested the hippocampal synaptic function itself. As for LTP in CA1 region, Qu et al. reported that LTP at the Schaffer-CA1 synapse in the hippocampal slice from PD model mice, which were generated by administration of MPTP (a dose of 30 mg/kg) intraperitoneally for 7 consecutive days, was significantly reduced compared to that from control mice [Qu et al., 2019, Aging 11, 1934-1964]. Therefore, we would really like to examine whether B. breve A1can improve the reduced LTP observed in MPTP-induced PD mice. Moreover, at the same time, we also need to test LTD changes in MPTP and B. breve A1 treatment groups because we have previously proposed an association between LTD and cognitive flexibility [Nishie et al., 2012, Neurosci. Res. 73, 292-301]. Although we know that it would take a little more time until we reach a conclusion, we will focus on both LTP and LTD at the Schaffer-CA1 synapse and analyze those changes in PD mice with and without administration of B. breve A1 to confirm the pharmacological effect of B. breve A1.
In response, we have added the sentences about this issue in the section of Discussion (page 11, line 393-401) and newly added Ref [37]. Page 11, line 393 now read “In the present study, we found that B. breve A1 restored the facilitation of contextual fear extinction, recovered the reduction of mRNA expression levels of two synaptic proteins, SYP and PSD95, and prevented the reduction of spine density. However, we have not yet tested the hippocampal synaptic function itself. Therefore, we need a further study to examine the effects of B. breve A1 on the long-term potentiation and the long-term depression, those of which are considered as neurophysiological substrates for learning and memory [37], at the Schaffer collateral–CA1 synapse in the hippocampus of PD mice.”
Reviewer 3 Report
This is an interesting article which evaluated the effects of oral administration of probiotic bifidobacterium breve on the improvement facilitation of hippocampal memory extinction in an MPTP-induced mouse model of Parkinson’s Disease. The Authors demonstrated that this probiotic restored abnormal hippocampal synaptic plasticity and fear extinction via restoration of aberrant higher induction of neuropsin. The manuscript is of high interest for scientific community, complete and well written. I have no major issues, but some points should be addressed:
- In the introduction, more recent studies and reviews should be added to better demonstrate the probiotic effects on the central nervous system via the microbiota–gut–brain axis (e.g Castelli et al, 2020 Aging; Perterson, 2020, Journal of Evidence-Based Integrative Medicine; Castelli et al, 2021 Neural Regeneration Research).
- In 2.1. Animals and MPTP, the Authors should specify the total number of animals used and for each group (n=x)
- In 2.5 Golgi–Cox staining, please clarify the microscopy used for images acquisition.
- In 2.7. Administration of B. breve, the Authors should clarify the type of administration used for the probiotics (e.g. oral Gavage)
- The representative western blotting figure of FIG8 should be replaced with a most representative one.
Author Response
Responses to reviewer #3 comments
Point 1: In the introduction, more recent studies and reviews should be added to better demonstrate the probiotic effects on the central nervous system via the micro-gut-brain axis (e.g. Castelli et al., 2020 Aging; Perterson, 2020, Journal of Evidence-Based integrative Medicine; Castelli et al., 2021 Neural Regeneration Research).
Response: We greatly appreciate your comments. In response, we have newly added the following reports, Perterson, C.T., J. Evid. Based. Integr. Med. 2020, 25, 1-19. as the number of [12], Castelli et al., Neural. Regen. Res. 2021, 16, 628-634. as the number of [13], and Castelli et al., Aging 2020, 12. 4641-4659. as the number of [15] in the list of References, and also revised and added the sentence about this issue in the section of Introduction (page 2, line 46-52). Page 2, line 46 now read “Furthermore, it has been reported that the application of some probiotics is beneficial for the central nervous system via the microbiota–gut–brain axis [11-13]. For example, administration of Bifidobactterium breve strain A1 [MCC1274] (B. breve A1) was shown to prevent cognitive impairment in Alzheimer’s disease (AD) model mice [14], daily administration of probiotic formulation SLAB51 was found to restore behavioral impairment and rescue DAergic neurons in SNpc and striatum in 6-hydroxydopamine -induced PD model mice [15], administration of Lactobacillus plantarum PS128 was found to improve anxiety-like behaviors in mice exposed to early life stress [16], and administration of VSL#3, a probiotic mixture of eight different gram-positive bacterial species, were found to modulate neuronal activities such as long-term potentiation in young and aged rats [17].”
Point 2: In 2.1. Animals and MPTP, the Authors should specify the total number of animals used and for each group (n=x).
Response: We greatly appreciate your comments. In response, we have added the sentences about the number of mice used and for each group in the section of Materials and Methods (page 2, line 78-81). Page 2, line 78 now read “Each of MPTP-treated (n = 73) and saline-treated mice (n = 73) was divided into the following three groups: Contro-Saline, n = 36; Control-B. breve A1, n = 32; Control-Non-viable B. breve A1, n = 5; MPTP-Saline, n = 36; MPTP-B. breve A1, n = 32; MPTP-Non-viable B. breve A1, n = 5.”
Point 3: In 2.5 Golgi-Cox staining, please clarify the microscopy used for images acquisition.
Response: We appreciate your comments. In response, we have newly added the sentence to clarify the microscopy used for images acquisition in the section of Materials and Methods (page 4, line 142-144). Page 4, line 142 now read “We analyzed the spine density of secondary apical dendrites in CA1 pyramidal neurons using biological light microscopy (BX-41, Olympus, Tokyo, Japan) and image capture system (DP-70 with DP manager, Olympus).”
Point 4: Administration of B. breve, the Authors should clarify the type of administration used for the probiotics (e.g. oral Gavage).
Response: We greatly appreciate your comments. In response, we have newly added the sentence to clarify the type of administration used for the probiotics in the section of Materials and Methods (page 4, line 179-181). Page 4, line 179 now read “For oral gavage, mice were administered using a mouse feeding stainless steel bulbous-ended needle (0.92 x 70 mm, AS ONE, Osaka, Japan) and the bulbous-ended needle was inserted over the tongue into the stomach.”
Point 5: The representative western blotting figure of FIG8 should be replaced with a most representative one.
Response: We greatly appreciate your comments. In response, we have replaced the image of western blotting with a most representative one in Figure 8.
Round 2
Reviewer 2 Report
The authors answered my concerns in the feedback.